# Sandwich Enzyme-Linked Immunosorbent Assay for Quantification of Callose

**DOI:** 10.3390/mps5040054

**Published:** 2022-06-26

**Authors:** Abubakar S. Mustafa, Jamilu E. Ssenku, Paul Ssemanda, Saidi Ntambi, Savithramma P. Dinesh-Kumar, Arthur K. Tugume

**Affiliations:** 1Department of Plant Sciences, Microbiology and Biotechnology, College of Natural Sciences, Makerere University, Kampala P.O. Box 7062, Uganda; mustafa.abubakar.sadik@gmail.com (A.S.M.); jssenku@gmail.com (J.E.S.); paulsmdd@gmail.com (P.S.); saidintambi08@gmail.com (S.N.); 2Department of Plant Biology, College of Biological Sciences, University of California, Davis, CA 95616, USA; spdineshkumar@ucdavis.edu

**Keywords:** callose, enzyme-linked immunosorbent assay (ELISA), *Xanthomonas campestris* pv. *musacearum*, banana, biotic stress

## Abstract

The existing methods of callose quantification include epifluorescence microscopy and fluorescence spectrophotometry of aniline blue-stained callose particles, immuno-fluorescence microscopy and indirect assessment of both callose synthase and β-(1,3)-glucanase enzyme activities. Some of these methods are laborious, time consuming, not callose-specific, biased and require high technical skills. Here, we describe a method of callose quantification based on Sandwich Enzyme-Linked Immunosorbent Assay (S-ELISA). Tissue culture-derived banana plantlets were inoculated with *Xanthomonas campestris* pv. *musacearum* (*Xcm*) bacteria as a biotic stress factor inducing callose production. Banana leaf, pseudostem and corm tissue samples were collected at 14 days post-inoculation (dpi) for callose quantification. Callose levels were significantly different in banana tissues of *Xcm*-inoculated and control groups except in the pseudostems of both banana genotypes. The method described here could be applied for the quantification of callose in different plant species with satisfactory level of specificity to callose, and reproducibility. Additionally, the use of 96-well plate makes this method suitable for high throughput callose quantification studies with minimal sampling and analysis biases. We provide step-by-step detailed descriptions of the method.

## 1. Introduction

Callose, a polysaccharide of β-1,3-glucan, occurs naturally in the cell walls of a variety of higher plants. Callose is synthesized by callose synthases (*CalS*) [1] and degraded by β-(1,3)-glucanases. It is naturally involved in numerous plant biological processes which include growth and development and response to abiotic and biotic stresses [2,3,4]. During pathogen infection, increased callose deposition in the papillae prevents further microbial colonization, acting as a permeability barrier between neighboring plant cells [2,5,6,7,8,9,10,11,12,13,14]. By slowing down pathogen invasion in the attacked tissue, callose deposition allows time for the induction of additional defense responses. The rate of callose deposition and subsequently quantity of callose accumulated in tissues or cell walls therefore has a strong bearing in physiological needs of the tissue at a given time [4,15].

The routinely used method of callose quantification involves imaging of aniline blue-stained callose particles by epifluorescence microscopy [6,16,17,18,19,20,21,22,23,24,25,26,27,28,29,30,31,32,33,34,35,36,37,38,39]. The aniline blue-stained callose under ‘blue’ or UV light excitation appears as yellow fluorescent particles, which are either followed by manual counting of the fluorescing callose particles [31,34] or by automated callose counting software. Manual counting of the fluorescing callose particles is laborious, time consuming and subjective [25], whereas the use of automated callose counting software requires acquisition of software resources and a considerate amount of technical skills. Some of the software used for automated callose counting include ImageJ [22,37,40], photoshop [23], CalloseMeasurer [16] and Icy [25,41]. The aniline blue-stained callose may also be quantified using fluorescence spectrophotometry [18,21,42,43]. However, the high background/autofluorescence associated with fluorescence spectrophotometry makes this method difficult and unreliable. Moreover, aniline blue can stain other β-1,3-glucans besides callose [44], which presents an enormous disadvantage of both epifluorescence microscopy and fluorescence spectrophotometry methods of callose detection and quantification. Consequently, these challenges make these two methods unreliable, non-user friendly and subjective for callose quantification.

A method of immunohistochemistry for callose quantification which is based on callose-specific antibodies has also been used in several studies [29,45,46]. Although this method has similar disadvantages to epifluorescence microscopy of aniline blue-stained callose, it has the advantage of being callose-specific due to the high callose-specific antibodies used. Assessment of callose synthases (*CalS*) [37,47] and β-(1,3)-glucanases activities [21,48] have also been used in many studies for indirect callose quantification. 

Here, we report a new method of callose quantification based on enzyme immunoassay, more specifically, Sandwich Enzyme-Linked Immunosorbent Assay (S-ELISA). The method can be optimized and applied for the quantification of callose in different plants or their tissues with satisfactory levels of specificity to callose, high precision and reproducibility. This method can also be modified to quantify any plant-based analyte if the antibody against that analyte is available. 

## 2. Experimental Design

### 2.1. Materials

*Xanthomonas campestris* pv. *musacearum* bacterial isolates, causative agent of banana Xanthomonas wilt (BXW) disease in banana.Two and half months-old tissue culture-derived *Musa balbisiana* and “Mbwazirume” banana plantlets. *Musa balbisiana* is a diploid (genome BB) wild progenitor of cultivated banana whereas “Mbwazirume” is triploid (genome AAA-EA) and a local commercial banana variety in Uganda belonging to the larger group of East African highland banana (EAHB) genotypes.Sodium hydroxide (NaOH) (PanReac AppliChem ITW Reagents—PanReac Química SLU, Barcelona, Spain; Cat. No.: 141687.1210).Sodium chloride (NaCl) (PanReac AppliChem ITW Reagents—PanReac Química SLU, Barcelona, Spain; Cat. No.: A2942).Potassium di-hydrogen phosphate (KH_2_PO_4_) (PanReac AppliChem ITW Reagents—PanReac Química SLU, Barcelona, Spain; Cat. No.: 141509).Di-sodium hydrogen phosphate (Na_2_HPO_4_) (PanReac AppliChem ITW Reagents—PanReac Química SLU, Barcelona, Spain; Cat. No.: 141679).Potassium chloride (KCl) (PanReac AppliChem ITW Reagents—PanReac Química SLU, Barcelona, Spain; Cat. No.: A2939).Sodium azide (NaN_3_) (PanReac AppliChem ITW Reagents—PanReac Química SLU, Barcelona, Spain; Cat. No.: A1430).Hydrochloric acid (HCl) (PanReac AppliChem ITW Reagents—PanReac Química SLU, Barcelona, Spain; Cat. No.: 141020).Sodium carbonate (Na_2_CO_3_) (PanReac AppliChem ITW Reagents—PanReac Química SLU, Barcelona, Spain; Cat. No.: 141648).Sodium hydrogen carbonate (NaHCO_3_) (PanReac AppliChem ITW Reagents—PanReac Química SLU, Barcelona, Spain; Cat. No.: 141638).Polyvinylpyrrolidone (PVP) (PanReac AppliChem ITW Reagents—PanReac Química SLU, Barcelona, Spain; Cat. No.: A2260).Diethanolamine (PanReac AppliChem ITW Reagents—PanReac Química SLU, Barcelona, Spain; Cat. No.: 191287).Micropipette tips (Eppendorf, Hamburg, Germany; Brand: epT.I.P.S. ^®^ Singles; Cat. No.: 022492209).Microcentrifuge tubes 1.5 mL (Eppendorf, Hamburg, Germany; Catalog No. 022363204).Microcentrifuge tubes 2 mL (Genesee Scientific Corp., San Diego, CA, USA; Cat. No.: 24–283).Reagent reservoirs (Thermo Fisher Scientific Inc., Waltham, MA, USA; Cat. No.: 15075).ELISA plates, 96-well, flat base, transparent, polystyrene, high binding (Sarstedt AG & Co. KG, Nümbrecht, Germany; Cat. No.: 82.1581.200).Laminarin (Alfa aesar, Haverhill, MA, USA; Cat. No.: J66193).Para-nitrophenyl phosphate (pNPP) (Merck KGaA, Darmstadt, Germany; Cat. No.: 20-106).Parafilm (Laboratory film) (Paul Marienfeld GmbH & Co. KG, Lauda-Königshofen, Germany; Cat. No.: 740751).Tween-20 (Biomatik Corporation, Ontario, Canada; Cat. No.: A4031).Bovine Serum Albumin (BSA) (Thermo Fisher Scientific, Waltham, MA, USA; Cat. No.: B14).Ultrapure distilled water (Thermo Fisher Scientific, Waltham, MA, USA; Cat. No.: 10-977-015).Blotting paper (Kim-Fay EA Limited, Nairobi, Kenya; Fay Kitchen towels).Aluminium foil (Kim-Fay EA Limited, Nairobi, Kenya).Primary antibody (1-3-β-glucan-directed mouse IgG) (Bio supplies Australia Pty Ltd., Melbourne, Australia; Cat. No.: 400-2).Primary antibody in coating buffer (see Reagents Setup).Primary antibody in blocking buffer (see Reagents Setup).Secondary antibody conjugated to Alkaline Phosphatase (anti-Mouse IgG-Alkaline phosphatase) (Sigma Life Sciences, Cherry Hill, NJ, USA; Cat. No.: A5153) (see Reagents Setup).Blocking buffer (see Reagents Setup).Coating buffer (see Reagents Setup).Wash buffer (see Reagents Setup).Conjugate buffer (see Reagents Setup).Substrate buffer (see Reagents Setup).Phosphate buffered saline (PBS) (see Reagents Setup).Para-nitrophenyl phosphate (pNPP) solution (see Reagents Setup).Microplate Manager^®^ 6 Version 6 software (Bio-Rad Laboratories, Inc., Hercules, CA, USA; Cat. No.: 1689520).

### 2.2. Equipment

Freeze-drier (LabWrench, Canada, USA; Brand: VirTis—BenchTop™ “K” series; Model: 4KBTXL; Cat. No.: 448053).Digital Shaker (Eppendorf, Hamburg, Germany; Brand: MixMate^®^; Cat. No.: 5353000529).Centrifuge (Eppendorf, Hamburg, Germany; Brand: 5425R; Cat. No.: 5406000240).Micropipettes (0.5–1000 μL) (Eppendorf, Hamburg, Germany; Brand: Research ^®^ plus; Cat. No.: 3123000900).12-channel micropipette (30–300 μL) (Eppendorf, Hamburg, Germany; Brand: 2100 series; Cat. No.: EP-12-300R).Microplate absorbance reader (Bio-Rad Laboratories, Inc., Hercules, CA, USA; iMark; Cat. No.: 1681135EDU).405 nm filter for the iMark microplate reader (Lasec International Pty. Ltd., Cape Town, South Africa; Model: iMark 680; Cat. No.: BRD1681011).Incubator (Esco Lifesciences Group, Singapore; Isotherm^®^ Forced Convection Incubator; Model: IFA-54-8, Cat. No.: 2100002).Combined refrigerator- freezer (4 °C and −20 °C) (Haier Medical and laboratory Co. Ltd., Qingda, China; Model: HYCD-282).Beadbeater 96 (BioSpec Products Inc., Bartlesville, OK, USA, Cat. No.: 1001EUR).Precision water bath (Thermo Fisher Scientific Inc., Newinton, CT, USA; Model: GP10; Cat. No.: TSGP10).Analytical balance (Mettler-Toledo AG, Greifensee, Switzerland; Model: ML204/01).pH meter (Hanna Instruments, Woonsocket, RI, USA; Model HI9126; S/N: 02310048991).

## 3. Procedure

### 3.1. Inoculation of Experimental Plants and Sampling

**OPTIONAL STEP** Inoculate 2.5-month-old tissue-culture-derived banana plantlets (*Musa balbisiana* and Mbwazirume) (Figure 1, Step 1) with 200 µL of the PCR-confirmed *Xcm* inoculum (1 × 10^8^ cells) on the dorsal side of the leaf petioles [49] (Figure 1, Step 2). Quantification of callose using this method can be done in any plant samples depending on the objectives of the study. In our case, we wanted to assess callose production in two banana genotypes infected with *Xcm* as a biotic stress imposed on the plants.**OPTIONAL STEP** Inoculate the control plantlets with 200 µL of double distilled sterile water. Replicate the experiments 6–10 times and repeat at least 2–3 times (Figure 1, Step 2).Quickly excise banana leaf, pseudostem and corm samples at 14 days post-inoculation (dpi) (time of sampling is variable depending on experimental design) and place them into labeled 50 mL falcon tubes (Figure 1, Step 3).

**! NOTE:** The 14 dpi was chosen as the sampling point because it has been shown to be the time at which the symptoms of bacterial wilt disease of banana appear following *Xcm* infection or inoculation [49,50].
4.Immediately immerse the 50 mL falcon tubes containing the samples into liquid nitrogen.5.Transport the samples and store at −80 °C.


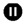
**PAUSE STEP** The samples can be stored at −80 °C for up to 1 year or longer with minimal freeze–thaw cycles. 

### 3.2. Sample Preparation (Time to Completion: 3 Days 2 h)

1. Freeze-dry the samples for 72 h using the Mini LYOTRAP freeze-drier (LTE Scientific Ltd., Greenfield, UK) (Figure 1, Step 4). 


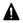
**CRITICAL STEP** The samples to be freeze-dried must never thaw and the Mini LYOTRAP freeze-drier must already have attained below 0 °C before placing the samples into it to avoid thawing the samples. 

2. Place 30 mg of the freeze-dried samples into 2 mL eppendorf tube containing 2 steel bicycle beads and pulverize the tissue samples into fine powder by the use of the Beadbeater 96 (BioSpec Products Inc., Bartlesville, OK, USA) (Figure 1, Step 5). 


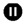
**PAUSE STEP** The pulverized samples can be stored in a cool dry environment at room temperature, well-sealed and safe from moisture for up to 4 months [51,52] or at −80 °C for up to 1 year with minimal freeze–thaw cycles. 

### 3.3. Extraction of Callose from the Banana Samples (Time to Completion: 1 h)

To the 30 mg of the powdered samples in 2 mL eppendorf tube, add 800 μL of 1 M NaOH and incubate for 30 min at 80 °C in a water bath, with occasional vortexing after every 10 min [43,53] (Figure 1, Step 6).Allow the tube to cool to room temperature (approximately 5 min) and centrifuge at 12,000 rpm for 5 min.**OPTIONAL STEP** Transfer the supernatant (callose extract) to a sterile 2 mL eppendorf tube and dilute the leaves, pseudostems and corms with the blocking buffer (see Reagents Setup) at a ratio of 1:1, 1:1 and 1:2, respectively. The callose extract may or may not be diluted. Once required, dilution should be done in the blocking buffer and the dilution ratio needs to be optimized for different sample types to get absorbance that lies within the range of the standard curve.


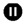
**PAUSE STEP** The callose extract can be stored at −20 °C for up to 6 months with minimal freeze–thaw cycles. Thaw to room temperature before use.

### 3.4. Preparation of Laminarin Standards and Blank (Time to Completion: 2 h)

Prepare a concentrated stock of laminarin standard at 100 mg/mL in 1 M NaOH [4,54,55,56,57].Incubate the resultant suspension of the standard at 80 °C in a water bath, with gentle shaking intervals, until all the laminarin dissolves (approximately 20–40 min).Cool the dissolved standard to room temperature (approximately 10 min).Using the blocking buffer (see Reagents Setup), prepare laminarin standards at concentrations of 80, 60, 40, 20, 10, 1, 0.5, 0.1 and 0.01 mg/mL from the concentrated stock of 100 mg/mL.Prepare the blank by mixing 1 M NaOH and the blocking buffer (see Reagents Setup) at a ratio of 1:2 (Use the dilution ratio used in Section 3.3, Step 3 above, for callose extract, if dilution is required).


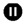
**PAUSE STEP** The laminarin standards and the blank can be stored at −20 °C for up to 6 months with minimal freeze–thawing cycles. Thaw to room temperature before use.

6. Use the callose extracts and standards to quantify callose in the plant tissues (Figure 1, Step 7–8) as indicated in Section 3.5 below. 

### 3.5. Quantification of Callose by S-ELISA (Time to Completion: 4 Days)

Add 100 μL of the primary antibody (1-3-β-glucan-directed mouse IgG) in a coating buffer (see Reagents Setup) to each of the wells of the plate (Figure 2, Step 1).Seal the plate tightly with parafilm and incubate overnight at 4 °C in a refrigerator.The next day, place the plate on the bench and allow it to get to room temperature (approximately 10 min).Wash the plate by standard blotting and washing procedures [58]. Briefly, add 200 μL of the wash buffer (see Reagents Setup) to the plate, vortex at 500 rpm for 30 sec using MixMate^®^ Digital Shaker (Eppendorf, Hamburg, Germany) and blot on tissue paper. Wash the plate 2 more times (Figure 2, Step 2).


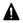
**CRITICAL STEP** Do not allow the plate to completely dry and use a sterile micropipette tip to remove any bubbles present in the wells without touching the base and walls of the plate.
5.Add 200 μL of the blocking buffer (see Reagents Setup) to each of the wells of the plate (Figure 2, Step 3).6.Seal the plate tightly with parafilm and incubate for 4 h at 37 °C.7.Repeat Step 3 and 4 (Figure 2, Step 4).8.Add 100 μL of the callose extracts obtained in Section 3.2 above to the designated wells of the plate (Figure 2, Step 5).9.Add 100 μL of the laminarin standards obtained in Section 3.4 above to the designated wells of the plate (Figure 2, Step 5).10.Add 100 µL of the blank obtained in Section 3.4 above to the designated wells of the plate (Figure 2, Step 5).11.Repeat Step 2 to 4 (Figure 2, Step 6).12.Add 100 µL of the primary antibody (1-3-β-glucan-directed mouse IgG) in blocking buffer (see Reagents Setup) to each of the wells of the plate (Figure 2, Step 7).13.Repeat Step 6, then Step 3 to 4 above (Figure 2, Step 8).14.Add 100 µL of the secondary antibody (anti-Mouse IgG-Alkaline phosphatase) (see Reagents Setup) to each of the wells of the plate [58,59] (Figure 2, Step 9).15.Repeat Step 2 to 4 above (Figure 2, Step 10).16.Add 100 µL of freshly prepared para-nitrophenyl phosphate (pNPP) solution at a concentration of 1 mg/mL (see Reagents Setup) to each of the wells of the plate (Figure 2, Step 11).17.**VARIABLE STEP** Incubate the plate on bench at room temperature for 30 min [60]. The incubation time may vary depending on the plant samples and standard used. In our case, laminarin had good readings between 20–40 min with best readings at 30 min.18.Terminate the reaction by addition of 100 µL of freshly prepared stop solution (0.5 M NaOH) to each of the wells of the plate [60].19.Transfer plate to iMark Microplate Reader (BIO-RAD, Japan) (equilibrated to 37 °C) and fitted with 405 nm filter, shake at medium speed for 1 min and read absorbances at 405 nm [60] (Figure 2, Step 12).

## 4. Expected Results

This protocol was developed to quantify callose in banana leaves, pseudostems and corms by the S-ELISA method that is based on callose-specific immunoglobulin G (IgG) and quantification of the callose by spectrophotometry. The leaf petioles of two and-one-half-month-old banana plantlets were inoculated with *Xcm* and leaf, pseudostem and corm samples were collected at 14 dpi, immediately frozen in liquid nitrogen and transported to the laboratory. The plant samples were then freeze-dried and pulverized into fine powder to increase the surface area of the plant tissues for callose extraction. Callose was extracted from the pulverized plant samples as described above. S-ELISA method was performed as described in Section 3. A simple linear regression of laminarin absorbance at 405 nm against laminarin concentration (µg/mL) was performed to obtain the standard curve, Y = β_1_X + β_0_ [(Y = laminarin absorbance at 405 nm, β_1_ = r = slope of the regression line = Pearson correlation coefficient, X = Log_10_ (laminarin concentration in µg/mL) and β_0_ = Y-intercept)]. The actual equation of the regression line is given as y = 0.5079x − 0.04534 and the laminarin standard curve is given in Mustafa et al. [50]. Callose absorbance and concentrations were therefore given as laminarin equivalents (LE). To compute callose concentrations in the plant samples, the absorbance of the blank was subtracted from the absorbance of the samples and the standard curve was used to estimate the callose concentration in µg/mL as LE. R statistical package, version 3.6.3 [61] was used to analyze all data. All data was checked for normality of distribution using the Shapiro–Wilk test and homoscedasticity of variances (α > 0.05). The independent sample *t*-test was used to compare the callose production between the *Xcm*-inoculated and control groups (*p* ≤ 0.05). 

Callose production in the leaves of *Xcm*-inoculated groups were statistically higher than the control groups in Mbwazirume (independent sample *t*-test, t_(12)_ = 4.9520, *p* < 0.001) and *Musa balbisiana* (independent sample *t*-test, t_(14)_ = 6.2617, *p* < 0.0001) (Figure 3, Appendix A). Similarly, callose production in the corms of *Xcm*-inoculated and control groups varied significantly in both the banana genotypes (independent sample *t*-test, *p* < 0.05). Contrary to the observation made for the leaves and corms, callose production in the pseudostems of *Xcm*-inoculated and control groups of both Mbwazirume and *Musa balbisiana* were not significantly different (independent sample *t*-test, *p* = 0.05142 and *p* = 0.05818, respectively) (Figure 3, Appendix A). Highest callose was produced in the corms of *Xcm*-inoculated Mbwazirume plantlets, whereas the lowest callose was produced in the leaves of *Musa balbisiana,* control group (Figure 3, Appendix A). This method was able to show consistency in our results. An example of the data generated by this method in comparison to the epifluorescence microscopy method is given in Mustafa et al. [50]. This method can be applied for the quantification of callose in different plant species with satisfactory level of specificity to callose, high precision and reproducibility. Moreover, use of 96-well plates makes this method suitable for high throughput callose quantification studies with minimal subjective sampling and analysis. This method of callose detection is as reliable as the immunofluorescence spectrophotometry method [29,45,46]. It is noteworthy that although callose may be easily extracted from freshly harvested plant tissues [18,43,45], the “sample preparation” step included in our protocol provides a leverage of keeping the samples for a very long time (up to 1 year) and allows shipping of perishable biological samples between distant laboratories at ambient temperatures. This allows opportunities of repeatability of callose extractions without the necessity of repeatedly setting up the entire experiment.

## 5. Reagents Setup

Phosphate-buffered saline (PBS), pH 7.4 (1 L) (store at −20 °C for up to 12 months). 8.0 g of sodium chloride (NaCl). 0.2 g of monobasic potassium phosphate (KH_2_PO_4_). 1.15 g of dibasic sodium phosphate (Na_2_HPO_4_). 0.2 g of potassium chloride (KCl). 0.2 g of sodium azide (NaN3). Dissolve in 900 mL of deionised H_2_O, adjust pH and make up to 1 L. Blocking buffer (1 L) (store at −20 °C for up to 6 months). 10 g of bovine serum albumin (BSA) (1% *w*/*v*).Dissolve in 900 mL of PBS, make up to 1 L with PBS.Coating buffer (pH 9.6) (1 L) (store at −20 °C for up to 12 months) 1.59 g of sodium carbonate (Na_2_CO_3_).2.93 g of sodium bicarbonate (NaHCO_3_).0.20 g of sodium azide (NaN_3_).Dissolve in 900 mL of deionised H_2_O, adjust pH and make up to 1 L.Primary antibody in coating buffer (store at −20 °C for up to 6 months) Reconstitute the lyophilized primary antibody according to manufacturer’s instruction to obtain the concentrated stock.Dilute the obtained stock in coating buffer to a working concentration of 2 μg/mL.Primary antibody in blocking buffer (store at −20 °C for up to 6 months) Reconstitute the lyophilized primary antibody according to manufacturer’s instruction to obtain the concentrated stock.Dilute the obtained stock in blocking buffer to a working concentration of 2 μg/mL.Wash buffer (1 L) (pH 7.5) (store at −20 °C for up to 12 months). 0.5 mL of Tween 20 (0.05% *v*/*v*).Dissolve in 990 mL of PBS, make up to 1 L with PBS.Conjugate buffer (1 L) (store at −20 °C for up to 6 months). 2 g of PVP (2% *w*/*v*).Dissolve in 900 mL of blocking buffer, make up to 1 L with blocking buffer.Secondary antibody (100 mL) (store at −20 °C for up to 6 months). 100 µL of secondary antibody concentrate.100,000 µL of conjugate buffer (1:1000 ratio).Substrate buffer (1 L) (pH 9.8) (store at −20 °C for up to 12 months). 97 mL of diethanolamine.600 mL of H_2_O.0.2 g of sodium azide (NaN_3_).Adjust pH 9.8 and make up to 1 L with H_2_O.Para-nitrophenyl phosphate (pNPP) solution (store at −20 °C for up to 12 months).

Dissolve para-nitrophenyl phosphate (pNPP) in substrate buffer to a working concentration of 1 mg/mL.

## Figures and Tables

**Figure 1 mps-05-00054-f001:**
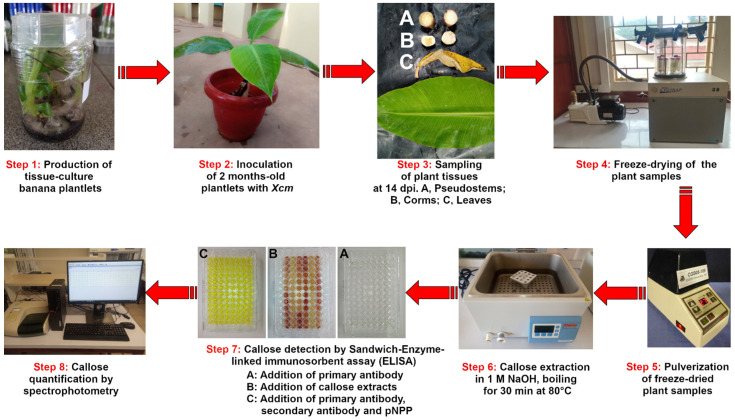
Schematic illustration of the major steps involved in callose quantification.

**Figure 2 mps-05-00054-f002:**
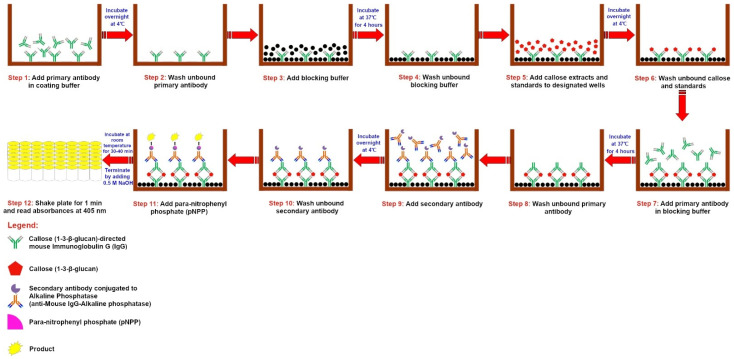
Schematic illustration of the Sandwich-Enzyme-Linked Immuno-Sorbent Assay (S-ELISA) for callose quantification.

**Figure 3 mps-05-00054-f003:**
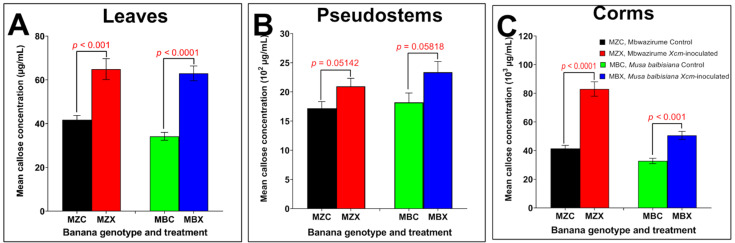
Mean callose concentration in the leaves (**A**), pseudostems (**B**) and corms (**C**) of *Xcm*-inoculated and control plantlets of *Musa balbisiana* and Mbwazirume (independent sample *t*-test, α ≤ 0.05). MZC = Mbwazirume control; MZX = Mbwazirume *Xcm*-inoculated; MBC = *Musa balbisiana* control; MBX = *Musa balbisiana Xcm*-inoculated.

## Data Availability

Not applicable.

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
