# Peer review of "Sandwich Enzyme-Linked Immunosorbent Assay for Quantification of Callose"

_mps, 2022, doi:10.3390/mps5040054_

Round 1

Reviewer 1 Report

The manuscript presents a clear method to detect and quantify callose. This has been always very challenging. Nevertheless, the quantification of callose can be made also with free software GIMP thus the "traditional" method for quantification can be affordable, if you have the proper microscope. With the TAS-ELISA you also need proper equipment and to acquire the antibodies, but it is true that the specificity can be improved with ELISA. 

In general, the method is well described and can be presented in the current form

Author Response

  1. Reviewer’s Comment:

The manuscript presents a clear method to detect and quantify callose. This has been always very challenging. Nevertheless, the quantification of callose can be made also with free software GIMP thus the "traditional" method for quantification can be affordable, if you have the proper microscope. With the TAS-ELISA you also need proper equipment and to acquire the antibodies, but it is true that the specificity can be improved with ELISA.

In general, the method is well described and can be presented in the current form

Authors’ Response/Action:

We thank this reviewer for this positive comments on our manuscript.

Reviewer 2 Report

The protocol by Mustafa et al describes a method to determine callose level in plant cell walls using a commercial monoclonal antibody in sandwich ELISA. The paper is well written and the topic of interest as callose regulates plant developmental and environmental responses.  Unfortunately there are some issues (in either the protocol or the description) that question the validity of the method for absolute callose quantification. Moreover there is no novelty in applying sandwich ELISA for detection of cell wall polysaccharides. This is a well-established method for cell wall glycans (see Cornuault and Knox, Bioprotocol, 2014). The only difference here is that the same callose antibody is used as capture and detection molecule which is in fact the major pitfall of the method. Specific comments are below, hoping these will help the authors to improve the manuscript.

1) line 19, the author rename the method Triple Antibody Sandwich ELISA. This is incorrect. There is no difference between the method described here and the well described sandwich ELISA cited above. The only difference is that only 1 primary antibody is used to capture and detection of callose, thus instead of 3, this is a two antibody approach: the anticallose and antimouse secondary.

2) line 25 "inoculated and control varied significantly" please rephrase. It should say ' callose level were significantly different in inoculated vs control groups'

3) line36: i would not say callose is abundant, it only represents 0.3% of Arabidopsis cell walls!

4) line 71: Antibodies are extremely expensive thus how is the method described here cheaper more 'attainable' than enzymatic method??

5) (section 3.3 and 3.4) Samples and standards are prepared in NaOH 1M, which is very high pH. This will degrade or at least denaturalize the antibody affecting their binding capacity. For this reason, it is difficult to believe this assay will work. The samples and standards should be neutralized before use as described in Cornuault and Knox.

6) line 270 Why is selecting only one standard a critical step? Would not be better having both? Please clarify in the text

7) the amount of antibody coating the plate is set thus this is a limiting factor for detection of callose (only a range of callose concentration can be detected with the antibody available in the plate). As it stands this method in only semiquantitative and requires to use a set of dilutions for all the samples (not only the standards) to account for this limitation.

8) When applying the secondary antibody this will detect both : the antibody that bind the callose on the top and the antibody that is coating the plate. Right? This should be clarified in FIG 2. This will allow for only semiquantitative assays (if the amount of antibody coating the plate is exactly the same) but the background (the blank) must being giving a very high signal, reducing the sensitivity of the assay. This is not clear in the explanation. What is the signal in the blanks?

9) antibodies have certain affinities and binding capacity. Thus they have lower and higher limits of detection, not linear in all concentrations as appear in fig.2 standards. It can not be assumed that the same formula can be used to calculate conc outside the standard concentrations. Since the authors indicate standard concentration used is between 0.7 and 80 mg/ml, it is difficult to see how the authors calculate concentrations in the order of 20ug/ml (leaves) and 4000mg/ml (corms).

10) the authors need to clarify the method is semiquantitative as absolute callose concentrations are not calculated per weight of cell wall but instead expressed in mg/ml. This is wrong as do not account for errors or different amount of starting cell wall/materials.

Author Response

  1.  Reviewer’s Comment:

The protocol by Mustafa et al describes a method to determine callose level in plant cell walls using a commercial monoclonal antibody in sandwich ELISA. The paper is well written and the topic of interest as callose regulates plant developmental and environmental responses.

Authors’ Response/Action:

We thank the reviewer for this positive comment on our manuscript.

  1. Reviewer’s Comment:

Unfortunately there are some issues (in either the protocol or the description) that question the validity of the method for absolute callose quantification. Moreover there is no novelty in applying sandwich ELISA for detection of cell wall polysaccharides. This is a well-established method for cell wall glycans (see Cornuault and Knox, Bioprotocol, 2014). The only difference here is that the same callose antibody is used as capture and detection molecule which is in fact the major pitfall of the method.

Authors’ Response/Action:

It is true that our method is relatively similar in principle to that of Cornuault and Knox (2014). However, there are significant differences between the two that make our method novel. The novelty of our protocol is in the specificity of the ELISA method in quantification of callose and not any other cell wall polysaccharides.

The protocol by Cornuault and Knox (2014) is non-specific and leads to the absolute quantification of all cell wall-derived glycans/polysaccharides. Although there is an option for extraction of only pectin using 50 mM cyclohexanediamine tetraacetic acid

(CDTA) (that would make their protocol pectin-specific), the primary monoclonal antibody specific to pectin is not provided.

The use of carbohydrate-binding module (CBM) as a capture molecule by Cornuault and Knox (2014) makes their protocol non-specific to a single component of the cell-wall glycans/polysaccharides since this molecule can bind to all the carbohydrates generated from the plant cells. On the contrary, our protocol design ensures that only callose is bound by the capture antibody (callose-specific), followed by its precise quantification by the detection antibody.

  1. Reviewer’s Comment:

Line 19, the author rename the method Triple Antibody Sandwich ELISA. This is incorrect. There is no difference between the method described here and the well described sandwich ELISA cited above. The only difference is that only 1 primary antibody is used to capture and detection of callose, thus instead of 3, this is a two antibody approach: the anticallose and antimouse secondary.

Authors’ Response/Action:

We appreciate the reviewer’s observation and have now changed the title of the manuscript to “Sandwich Enzyme-Linked Immunosorbent Assay for Quantification of Callose”. Similar changes have been made throughout the document.

  1. Reviewer’s Comment:

Line 25 "inoculated and control varied significantly" please rephrase. It should say ' callose level were significantly different in inoculated vs control groups'

Authors’ Response/Action:

As per reviewer’s suggestion, we have modified the sentence to read “Callose production levels were significantly different in banana tissues of Xcm-inoculated and control groups”.

  1. Reviewer’s Comment:

Line36: I would not say callose is abundant, it only represents 0.3% of Arabidopsis cell walls!

Authors’ Response/Action:

This has been corrected by deleting the statement “and is abundant”.

  1. Reviewer’s Comment:

Line 71: Antibodies are extremely expensive thus how is the method described here cheaper more 'attainable' than enzymatic method??

Authors’ Response/Action:

This has been corrected by deleting the statement “Though quite robust, these enzyme-based indirect callose quantification methods are expensive making them unattainable”.

  1. Reviewer’s Comment:

Section 3.3 and 3.4 Samples and standards are prepared in NaOH 1M, which is very high pH. This will degrade or at least denaturalize the antibody affecting their binding capacity. For this reason, it is difficult to believe this assay will work. The samples and standards should be neutralized before use as described in Cornuault and Knox.

Authors’ Response/Action:

We used 1M NaOH which has a pH of approximately 13 to make the stock of 100 mg/mL of the laminarin standard. This 100 mg/mL of standard stock was used to make the lower concentrations of the standards by diluting with blocking buffer [1% w/v bovine serum albumin (BSA) in phosphate buffered saline (PBS), pH 7.4]. The callose extracts were diluted in the blocking buffer at a ratio of 1:2 for the leaves and pseudostems and 1:3 for the corms before callose quantification by ELISA. Dilution of the standards and the callose extracts with the blocking buffer (pH 7.4) led to lowering of the pH significantly, hence no harm to the binding capacity of the antibodies or possibly, denaturation.

Furthermore, Cornuault and Knox (2014) use 4M KOH and neutralize with 1% (w/v) NaBH4. In our protocol, we use 1M NaOH which is four times lower in molar strength than 4M KOH. Preliminary studies showed that the protocol works by having approximately linear laminarin and callose absorbances with increasing concentrations.  

Extraction of callose using 1M NaOH has also been reported elsewhere (Köhle et al., 1985; Kohler et al., 2000; Khaledi et al., 2018). 

  1. Reviewer’s Comment:

Line 270 Why is selecting only one standard a critical step? Would not be better having both? Please clarify in the text

Authors’ Response/Action:

This has been corrected. Since the pachyman standard was not optimized to run through the range of the sample, we deleted it from the manuscript. Laminarin standard was optimized to accommodate the limits of the sample detection for all tissues.

  1. Reviewer’s Comment:

The amount of antibody coating the plate is set thus this is a limiting factor for detection of callose (only a range of callose concentration can be detected with the antibody available in the plate). As it stands this method in only semiquantitative and requires to use a set of dilutions for all the samples (not only the standards) to account for this limitation.

Authors’ Response/Action:

We were aware of this limitation right from the start of protocol development. In our preliminary studies we optimized and maximized callose quantification by trying out several dilutions of the standards, samples and blanks. This is the reason why low plant material (30 mg of plant tissue in 800 µL of 1M NaOH) was used in callose extraction and the callose extracts were diluted in the blocking buffer at a ratio of 1:1 for the leaves and pseudostems and 1:2 for the corms before callose quantification by ELISA. This was to ensure the callose concentration in the samples were low enough to maximize binding of callose to the primary antibody and optimal callose detection.

  1. Reviewer’s Comment:

When applying the secondary antibody this will detect both: the antibody that bind the callose on the top and the antibody that is coating the plate. Right? This should be clarified in FIG 2. This will allow for only semiquantitative assays (if the amount of antibody coating the plate is exactly the same) but the background (the blank) must being giving a very high signal, reducing the sensitivity of the assay. This is not clear in the explanation. What is the signal in the blanks?

Authors’ Response/Action:

The possibility that the secondary antibody could bind to the primary antibody cannot be ruled out as is the case in other forms of ELISA-based assays; however, ensuring that effective blanking is a good strategy. The blanks had very low signal in this method,  and the final absorbances considered for calculations were the difference between the signal detected for a sample/standard and the average signal detected for blanks on the same plate/reading.

  1. Reviewer’s Comment:

Antibodies have certain affinities and binding capacity. Thus they have lower and higher limits of detection, not linear in all concentrations as appear in fig.2 standards. It can not be assumed that the same formula can be used to calculate conc outside the standard concentrations. Since the authors indicate standard concentration used is between 0.7 and 80 mg/ml, it is difficult to see how the authors calculate concentrations in the order of 20ug/ml (leaves) and 4000mg/ml (corms).

Authors’ Response/Action:

We thank the reviewer for pointing this out. The assumption of linearity is only possible within the boundaries of the standard used. We realized that the reported concentrations were too high to our expectation as observed by Reviewer #2 and we had to critically assess the entire data.

Through this peer-review process, we realized that we had provided the wrong standard curve. During the preliminary experimentation, we had used the current Fig. 3A as our standard curve for callose quantification but we realized that the callose quantities in the leaves were too low while the callose quantities in the corms were too high falling outside the standard curve as observed by Reviewer #2. We later on came up with a new standard curve that runs from 0.01 mg/mL to 80 mg/mL. The correct standard curve has been provided that was used in (Mustafa et al., 2022). The old figure 3 from the previous version of the manuscript has been removed and replaced with new figure 3 (this is modified version of figure 4 in the previous version).

Though the absorbance we report fall within the standard curve, we had made an error by multiplying the absorbance with the dilution factor instead of multiplying the computed concentrations with the dilution factor. This is the reason why the concentrations went beyond the limits of the standard curve. We have corrected the entire data set by first computing the callose concentrations from the absorbance readings using the equation y = 0.5079x - 0.04534 followed by multiplication with the dilution factor. The callose extracts from the leaves, pseudostems and corms were diluted by the blocking buffer in the ratio of 1:1, 1:1 and 1:2, respectively. Due to these changes, figure 4 (now figure 3) and Supplementary Table S1 have been corrected.

Since this optimization was not performed with pachyman, the standard curve for pachyman was deleted.

  1. Reviewer’s Comment:

The authors need to clarify the method is semiquantitative as absolute callose concentrations are not calculated per weight of cell wall but instead expressed in mg/ml. This is wrong as do not account for errors or different amount of starting cell wall/materials.

Authors’ Response/Action:

It is true the method is semiquantitaive. It gives callose quantities based on our starting material, which was standardized across all samples. However the same starting mass may not be representative of the cell wall mass to render the quantification more absolute.

This method is indeed callose specific and semiquantitative in nature. The callose concentrations here are presented as µg/mL of the callose extract (cell wall material) but can as well be presented as µg/mg or µg/g of plant tissue since we know the amount of starting material (30 mg).

Reviewer 3 Report

In this manuscript, the authors describe a protocol for callose quantification based on Triple Antibody Sandwich Enzyme-Linked Immuno-sorbent Assay (TAS-ELISA).
The protocol is easy to follow and well described. The required equipment, reagents, etc. are well documented. Statistical analysesare simple and clear. figures are clear and easy to read.

I do not have major objections.

minor comments:
Lines 17-18: "These methods are laborious, time consuming, not
callose-specific, biased and require high technical skills."
Please specify which methods are not callose specific; aniline blue staining, yes, but immunofluorescence microscopy is callose specific.
You write that later on in the text, but here all the techniques were collapsed together.

In addition, change time consuming to time-consuming.

-In figure 1, step 7, change seconday antibody to a secondary antibody.

-Could you elaborate on why "This method of callose detection is more reliable than the immunofluorescence spectrophotometry method" as you write on line 345.

-Also, I would not agree that your method is less technically demanding, than immunodetection of callose. It is just complementary.

-When using NaOH to extract callose, are you also extracting other polysaccharides?

-would you say that your approach is the most sensitive to detecting callose changes between different samples?

Author Response

  1. Reviewer’s Comment:

In this manuscript, the authors describe a protocol for callose quantification based on Triple Antibody Sandwich Enzyme-Linked Immuno-sorbent Assay (TAS-ELISA). The protocol is easy to follow and well described. The required equipment, reagents, etc. are well documented. Statistical analyses are simple and clear. Figures are clear and easy to read.

I do not have major objections.

Authors’ Response/Action:

We thank the reviewer for these positive comments.

  1. Reviewer’s Comment:

Lines 17-18: "These methods are laborious, time consuming, not callose-specific, biased and require high technical skills. “Please specify which methods are not callose specific; aniline blue staining, yes, but immunofluorescence microscopy is callose specific. You write that later on in the text, but here all the techniques were collapsed together. In addition, change time consuming to time-consuming.

Authors’ Response/Action:

This has been corrected.

  1. Reviewer’s Comment:

In figure 1, step 7, change seconday antibody to a secondary antibody.

Authors’ Response/Action:

This has been corrected on Figure 1

  1. Reviewer’s Comment:

Could you elaborate on why "This method of callose detection is more reliable than the immunofluorescence spectrophotometry method" as you write on line 345.

Authors’ Response/Action:

This has been corrected to read “This method of callose detection is as reliable as the immunofluorescence spectrophotometry method”

  1. Reviewer’s Comment:

Also, I would not agree that your method is less technically demanding, than immunodetection of callose. It is just complementary.

Authors’ Response/Action:

This has been corrected

  1. Reviewer’s Comment:

When using NaOH to extract callose, are you also extracting other polysaccharides?

Authors’ Response/Action:

Yes. The 1 M NaOH is used to extract all the cell wall polysaccharides but the ELISA method is callose specific since callose-specific monoclonal antibodies are used under this method.

  1. Reviewer’s Comment:

Would you say that your approach is the most sensitive to detecting callose changes between different samples?

Authors’ Response/Action:

We would say that our method may not be the most sensitive. However, its sensitivity is comparable to immunofluorescence microscopy method as both of them are callose-specific unlike the standard gold method of aniline-blue staining which could easily detect other glycans of structural similarity to callose. The extraction of callose under our method allows a quantitative assessment of callose deposition in tissues while immunofluorescence microscopy allow histolocalization. The two methods are for example complementary depending on the needs.

This manuscript is a resubmission of an earlier submission. The following is a list of the peer review reports and author responses from that submission.